# Factors Affecting BMI Changes in Mothers during the First Year Postpartum

**DOI:** 10.3390/nu15061364

**Published:** 2023-03-11

**Authors:** Alissa D. Smethers, Jillian C. Trabulsi, Virginia A. Stallings, Mia A. Papas, Julie A. Mennella

**Affiliations:** 1Monell Chemical Senses Center, Philadelphia, PA 19104, USA; 2Department of Behavioral Health and Nutrition, University of Delaware, Newark, DE 19173, USA; 3Children’s Hospital of Philadelphia and Department of Pediatrics, Perelman School of Medicine, University of Pennsylvania, Philadelphia, PA 19104, USA; 4Institute for Research on Equity and Community Health, ChristianaCare, Wilmington, DE 19801, USA

**Keywords:** lactation, breast feeding, formula feeding, postpartum period, weight loss, eating behavior, prepregnancy body mass index

## Abstract

We tested the hypotheses that mothers of infants who exclusively breastfed would differ in the trajectories of postpartum BMI changes than mothers of infants who exclusively formula fed, but such benefits would differ based on the maternal BMI status prepregnancy (primary hypothesis) and that psychological eating behavior traits would have independent effects on postpartum BMI changes (secondary hypothesis). To these aims, linear mixed-effects models analyzed measured anthropometric data collected monthly from 0.5 month (baseline) to 1 year postpartum from two groups of mothers distinct in infant feeding modality (Lactating vs. Non-lactating). While infant feeding modality group and prepregnancy BMI status had independent effects on postpartum BMI changes, the benefits of lactation on BMI changes differed based on prepregnancy BMI. When compared to lactating women, initial rates of BMI loss were significantly slower in the non-lactating women who were with Prepregnancy Healthy Weight (β = 0.63 percent BMI change, 95% CI: 0.19, 1.06) and with Prepregnancy Overweight (β = 2.10 percent BMI change, 95% CI: 1.16, 3.03); the difference was only a trend for those in the Prepregnancy Obesity group (β = 0.60 percent BMI change, 95% CI: −0.03, 1.23). For those with Prepregnancy Overweight, a greater percentage of non-lactating mothers (47%) gained ≥ 3 BMI units by 1 year postpartum than did lactating mothers (9%; *p* < 0.04). Psychological eating behavior traits of higher dietary restraint, higher disinhibition, and lower susceptibility to hunger were associated with greater BMI loss. In conclusion, while there are myriad advantages to lactation, including greater initial rates of postpartum weight loss regardless of prepregnancy BMI, mothers who were with overweight prior to the pregnancy experienced substantially greater loss if they breastfed their infants. Individual differences in psychological eating behavior traits hold promise as modifiable targets for postpartum weight management.

## 1. Introduction

Gaining too much weight during pregnancy, and then retaining that weight, is not only distressing for women but also increases health risks both for themselves and for their children [1,2]. Beginning in the second trimester, women need to ingest additional energy to support optimal growth and maintenance of the fetus, placenta, and maternal tissues [3]. To reduce risks of maternal–fetal and neonatal complications, the 2009 Institute of Medicine and National Research Council (IOM/NRC) issued recommendations for optimal gestational weight gain (GWG) based, in part, on their prepregnancy body mass index (BMI) [4].

After parturition, energy needs differ based on how mothers feed their infants, since lactation imposes increased energy costs [3,5]. The consensus on the effects of lactation on postpartum weight loss remains inconclusive, however. Some studies [6,7,8,9], but not all [10,11,12], have reported that lactating women lost more body weight by 6 months postpartum than did non-lactating women. In addition to the infant feeding modality, a variety of biological and socioeconomic factors [8] have been shown to contribute to whether weight is retained or lost during the postpartum period. For example, women who retained more than 20 pounds at 1 year postpartum were more likely to have prepregnancy overweight or obesity; to identify as Black Americans; to be younger, poor, and on public assistance; and to have received high school or less education [8].

The present study had two goals. The primary goal was to test the hypothesis that mothers of infants who were exclusively breastfed would lose more weight postpartum than mothers of infants who were exclusively formula fed, but such benefits would differ based on the maternal BMI status prepregnancy. Rather than calculating a dose of breastfeeding based on its duration and exclusivity over time [7], we focused on two distinct groups with no overlap in breastfeeding dosage. One group of mothers exclusively formula fed and never breastfed their infants (hereafter referred to as the Non-lactating group) whereas the other group breastfed their infants for at least 4.5 months with no or little formula feeding (hereafter referred to as the Lactating group). The secondary goal was to determine whether baseline differences in eating behavior traits [13], when added to the model as fixed effects, predicted differences in the trajectory of BMI changes during the first year postpartum. A recent weight-loss intervention trial revealed that dietary restraint predicted weight loss in women with overweight or obesity [14].

## 2. Materials and Methods

### 2.1. Participants and Trial Designs

We conducted secondary analyses on anthropometric data collected from two groups of mothers whose infants had participated in one of two experimental, longitudinal trials conducted in Philadelphia, Pennsylvania [15,16]. For both trials, we enrolled women who recently gave birth to term infants but excluded those who had diabetes, gestational diabetes, or a family history of atopy. The study designs and other inclusion and exclusion criteria were published previously [15,16]. The trials were conducted according to the guidelines of the Declaration of Helsinki, approved by the University of Pennsylvania Institutional Review Board, and registered online at clinicaltrials.gov prior to their start.

By design, the trials recruited and selected two distinct groups of recently parturient women who differed in infant feeding modality. One trial was comprised of lactating women (the Lactating group) and was designed to determine the effects of maternal diet on children’s vegetable acceptance (NCT01667549; 2012–2015) [16]. These mothers breastfed their infants and fed no or small amounts of formula during the first 4.5 months, and the majority (81%) were lactating at 10.5 months, and then 1 year (62%) postpartum. The other trial, comprised of mothers whose infants were exclusively formula fed (the Non-lactating group), determined the effects of formula composition on children’s growth and energy balance (NCT01700205; 2012–2016) [15]. The infants in this group were never breastfed. Throughout the trials, study personnel did not comment on the women’s body weight.

### 2.2. Prepregnancy BMI Category Groupings

Because we predicted that prior body weight status was an effect modifier on postpartum weight loss, we calculated their prepregnancy BMI (kg/m^2^) from self-reported prepregnancy weight and measured height and formed the following groups: Prepregnancy Healthy Weight (BMI, 18.5–24.9 kg/m^2^); Prepregnancy Overweight (BMI, 25.0–29.9 kg/m^2^); and Prepregnancy Obesity (BMI ≥ 30.0 kg/m^2^). There were insufficient numbers to form the prepregnancy underweight group, these women (*n* = 6) were grouped with those having a prepregnancy healthy weight (*n* = 95); thus, herein “healthy weight” refers to BMI ≤ 24.9 kg/m^2^. In line with people-first language, “with obesity” was used herein instead of “obese” [17,18].

Recalled body weights were used to calculate prepregnancy BMI. To assess reliability, we correlated each woman’s prepregnancy BMI with their BMI at 0.5 month postpartum (which was based on measured body weights and heights) and found they were highly correlated (r = 0.93). The strength of such correlations indicates the reliability of the recalled prepregnancy body weight data [2]. At 0.5 month postpartum (baseline), study participants self-identified their race and completed questionnaires to collect data on various attributes (e.g., age, household income, education, parity, GWG), established a priori [8,19], that have been shown to impact postpartum weight loss and thus were identified as potential confounders in the analyses.

### 2.3. Anthropometry

Beginning at baseline and at every study visit thereafter, trained research personnel measured participants’ weight (kg) and height (m) in duplicate while participants wore light clothing and no shoes, from which BMI was calculated. Height and weight measurements for the Non-lactating group were obtained 13 times (time = 0.5, 1.5, 2.5, 3.5, 4.5, 5.5, 6.5, 7.5, 8.5, 9.5, 10.5, 11.5, 12.5 months), and for the Lactating group, eight times (time = 0.5, 1.5, 2.5, 3.5, 4.5, 7.5, 10.5, 12.5 months). Herein we refer to 12.5 months postpartum as the 1 year time point. At each time point, BMI was transformed into percent change from baseline (0.5 month), defined as ((post-baseline BMI (time)—baseline BMI)/baseline BMI) × 100. We made imputations for 17 missing data points due to missed visits by taking the mean of the two months surrounding the missing month or carrying the data from the prior month forward when it was the last visit. From these data, we determined the primary outcome measure of changes in maternal anthropometry, defined as percent changes in BMI relative to the start of the postpartum period; the percent change at 0.5 months was zero for each mother. These data were the dependent variables in the analyses. Because maternal height did not change during the postpartum period, if the percent BMI change decreased, then body weight decreased (weight losses), and if percent BMI change increased, then body weight increased (weight gains). We also determined how close in BMI the women were at 1 year postpartum to their prepregnancy BMI by subtracting BMI at 1 year from prepregnancy BMI.

We focused on percent changes in BMI, rather than changes in body weight (kg) for two reasons. First, comparisons of BMI change minimize bias because they take into account the wide variation in height among women [20]. Second, the amount and rate of weight loss during a specified unit of time depends on how close individuals are to their steady-state weight. Using a mathematical modeling approach, Hall and colleagues [21] demonstrated that those with greater adiposity have more weight to lose and the duration to reach their steady-state weight will be longer. Thus, comparing the percent BMI change over time, rather than changes in bodyweight per se, among postpartum women minimizes bias and allows for comparisons, especially among groups who differ in BMI at baseline [14,22].

### 2.4. Psychological Eating Behavior Traits

The secondary goal focused on the psychological eating-behavior traits of dietary restraint, disinhibition, and susceptibility to hunger that were measured at the start of the postpartum period. Each woman independently completed the Three-Factor Eating Questionnaire (TFEQ) developed by Stunkard and Messick [13], on computers equipped with Compusense™ 5 Plus (Guelph, Ontario, Canada). This 51-item, true–false questionnaire yielded three separate continuous, cognitive constructs: (a) dietary restraint, defined as a tendency to consciously restrict or control food intake, regardless of physiological signs of hunger and satiety (e.g., “I consciously hold back at meals in order not to gain weight”); (b) disinhibition, defined as a tendency to overeat relative to physiologic need and feeling of lack of control (e.g., “Sometimes when I start eating, I just can’t seem to stop”); and (c) susceptibility to hunger, defined as one’s food intake in response to perceived physiological symptoms of hunger (e.g., “I often feel so hungry that I just have to eat something”) [23]. The internal consistency coefficients (Cronbach’s *α*) of each construct were acceptable: 0.84 for restraint, 0.77 for disinhibition, and 0.73 for susceptibility to hunger.

### 2.5. Statistical Analyses

We took several approaches to test the primary hypothesis that the mother’s body weight status prior to pregnancy would interact with whether or not she lactated in predicting the direction and trajectories of BMI changes during the first year postpartum. First, we fit a linear mixed-effects model to explore the independent and interactive effects of prepregnancy BMI (Healthy Weight, Overweight, Obesity groups) and infant feeding modality (Lactating, Non-lactating groups), using the Prepregnancy Healthy Weight group and the Lactating group as reference groups. Mixed-effects linear models were used because they are a statistical technique that can account for unbalanced data patterns. Second, three separate linear mixed-effects models examined the effect of not lactating (i.e., exclusive formula feeding) for each prepregnancy BMI group, using the Lactating group as the reference group in each model.

All linear mixed-effects models included a random intercept for each mother and a random slope for time (measured in monthly intervals from 0.5 month to 1 y) to fit a trajectory that examined percent BMI change over 1 y. Time (months postpartum) was treated as a continuous variable, with polynomial factors of time tested and included if they improved model fit. The model with the lowest Bayesian information criterion was considered the best fit. The linear coefficient time represents the rate of BMI change (per month), and the quadratic coefficient time^2^ represents the acceleration or deceleration of the BMI change (per month^2^). Next, we tested the baseline characteristics (that were determined a priori as potential confounders), for inclusion in the models by examining the bivariate relationships of each with the dependent variable of percent BMI change. Parity did not meet the inclusion criterion as it was not associated with the dependent variable. Then, we examined the relationship among the potential confounders and found income, race, and education to be highly correlated, so we included income only to avoid multicollinearity. The fixed effects in the models were (pre-pregnancy BMI group, infant feeding modality group) time, time^2^, age, prepregnancy weight gain, and income. After comparison of models using potential correlation matrices, an unstructured correlation matrix was chosen. The models were run both with and without the 17 imputations, and the findings remained the same.

To test the secondary goal, the baseline measures of the psychological constructs of dietary restraint, disinhibition, and susceptibility to hunger (which were continuous variables) were independently added to the model as fixed effects in a progressive manner to determine whether they predicted postpartum BMI trajectories. We fit three mixed-effects linear models: Model 1 established the reference model for testing the three TFEQ constructs; Model 2 tested the interactions of each construct independently with time; and Model 3 added the interactions between each independent construct and polynomial factor of time (time^2^). All three models included the prepregnancy BMI group, the infant feeding modality group, the linear and quadratic variables of time, and baseline characteristics.

Descriptive statistics examined the distribution of baseline characteristics of age, race, income, education, parity, GWG, and prepregnancy anthropometric measures by infant feeding modality. We used two-sample *t*-tests for continuous variables and chi-squared tests for categorical variables. Marginal means and 95% confidence intervals (CI) generated from the linear mixed-effect models were used to describe the percent change in BMI from baseline. Outcomes from statistical models are displayed as mean ± standard deviation (SD) unless otherwise stated. Significant interactions were followed up by splitting the variables into meaningful groups to illustrate findings. Significant differences were determined at α level of 0.05. All analyses were performed using Stata/IC 17 (College Station, TX, USA) and Statistica version 14.1 (Tulsa, OK, USA).

## 3. Results

### 3.1. Participant Characteristics

Table 1 shows the baseline characteristics of the study population.

Total enrollment for the two trials was 210 women, with 159 (76%) completing the 1 year postpartum visit (Figure A1). However, no anthropometric data were available for two women so our final sample size was 208 (96 Lactating, 112 Non-lactating), unless otherwise indicated. Based on trial enrollments, 46% were lactating mothers and 54% were non-lactating mothers. Income data were missing for two and TFEQ data were missing for seven mothers.

The participants were diverse in racial self-identification, household income, and education level, which reflected the diversity of the urban setting in which they lived [24]. Of the 159 women who completed the 1 year visit (77 Lactating, 81 Non-lactating), anthropometric data were available for all but one mother. Half of the women gained more weight during pregnancy than was recommended by the IOM/NRC [4]. Compared to the Lactating group, a greater percentage of the women in the Non-lactating group were multiparous and identified as Black, whereas a smaller percentage in the Non-lactating group identified as White. Women in the Non-lactating group were significantly younger, had lower household incomes, had attained lower education levels, were less likely to have a healthy weight prepregnancy, and were less likely to meet IOM/NRC’s GWG recommendations. Among the Lactating group, the proportion who were still breastfeeding at 1 year did not differ by prepregnancy BMI group (Prepregnancy Healthy Weight group (64%) vs. Prepregnancy Overweight group (55%) vs. Prepregnancy Obesity group (50%), *p* = 0.57).

### 3.2. Independent and Interactive Effects of Infant Feeding Modality and Prepregnancy BMI

BMI trajectories were significantly and independently associated with infant feeding modality (*p* < 0.001) and prepregnancy BMI (*p* < 0.001). Three-way interactions were significant for (a) prepregnancy BMI group × infant feeding modality group × time and for (b) prepregnancy BMI group × infant feeding modality group × time^2^: the output of the three-way interaction model is presented in Appendix A (Appendix A).

To explore significant three-way interactions with time and time^2^, we examined each prepregnancy BMI group separately to determine whether the BMI trajectories of non-lactating women differed from those of lactating women (primary hypothesis). Table 2 summarizes the beta coefficient and 95% CI generated from the three models.

Figure 1A illustrates the BMI trajectories during the postpartum period for the Prepregnancy Healthy Weight group. In this group, compared to lactating women, the non-lactating women initially had significantly slower losses in BMI (0.63 change/month, 95% CI: 0.19, 1.06) and greater deceleration (−0.05 change/month^2^, 95% CI: −0.07, −0.02). The average BMI of the non-lactating women at 1 year was 7.3 ± 9.5% lower (mean loss of 4.4 ± 6.2 kg) than at baseline, which was not significantly different from the 8.4 ± 7.5% loss (mean loss of 5.7 ± 5.3 kg) for the lactating women.

As a group, women in the Prepregnancy Healthy Weight group experienced 7.4 ± 8.7% BMI loss from 0.5 month to 1 y. In the Prepregnancy Overweight group (Figure 1B), the non-lactating women had significantly greater gains in BMI (2.10 change/month, 95% CI: 1.16, 3.03) and greater deceleration (−0.11 change/month^2^, 95% CI: −0.17, −0.06) compared to the lactating women. At 1 y, for non-lactating mothers, BMI was 2.1 ± 13.0% higher than baseline (mean gain, 1.6 ± 10.5 kg), whereas for lactating mothers it was 6.3 ± 6.6% lower (mean loss, 4.1 ± 7.2 kg; *p* = 0.08).

In the Prepregnancy Obesity group (Figure 1C), while there was a tendency for non-lactating women to experience greater gains in BMI trajectories (0.60 change/month, 95% CI: −0.03, 1.23), there was no significant interaction with time^2^ (−0.03 change/month^2^, 95% CI: −0.06, 0.01). At 1 y, the average BMI change was not significantly different between non-lactating (4.4 ± 1.5%; mean gain, 3.5 ± 10.0 kg) and lactating (1.6 ± 2.0%; mean gain, 2.7 ± 10.9 kg) women. Overall, the Prepregnancy Obesity group had a 3.5 ± 9.6% BMI gain by 1 y.

As shown in Table 2, the influence of income and GWG depended on prepregnancy BMI status and socioeconomic status at the time of enrolment. In the Prepregnancy Healthy Weight and Obesity groups, the greater the GWG, the greater the BMI loss at 1 y. In contrast, in the Prepregnancy Overweight group, higher incomes were associated with a greater percent BMI loss.

### 3.3. BMI at 1 year Postpartum Relative to Prepregnancy

When we focused on the difference between their prepregnancy BMI and BMI at 1 year postpartum, we found that, as a whole, the women ranged from being 5.6 BMI units lower to 15.6 units higher (mean: 1.5; median: 1.4) than they were prior to the pregnancy. Women in the 25th quartile for percent BMI change lost 1.5 BMI units (range, −5.6–0 units; referred to as ≤0 units); those in the 25th–75th percentile gained 1.4 units (>0–2.9 units); and those in the 75th quartile gained 5.8 units (3.0–15.6 units; referred to as ≥3 units). At 1 y, the mean difference in BMI units between the upper and lower quartile of participants (<25th vs. >75th percentile: −1.5 vs. 5.8 units) was 7.3, indicating a substantial variability.

While the BMI trajectories differed between lactating and non-lactating women in the Prepregnancy Healthy Weight group, there were no group differences either in the absolute difference in BMI at 1 year relative to prepregnancy (mean, 1.0 units; 95% CI: 0.4, 1.6) or in the percentile categorization (Figure 2A). Neither were there differences between lactating and non-lactating women in the Prepregnancy Obesity group in absolute BMI units (mean, 2.2 units; 95% CI: 0.5, 3.5) or in the percentile categorization (Figure 2C). In contrast, in the Prepregnancy Overweight group, non-lactating women significantly differed from lactating women (Figure 2B): 47.1% of non-lactating women were ≥3 BMI units heavier at 1 y, compared to 9.1% of lactating women.

### 3.4. The Effects of the Psychological Eating Behavior Traits

The outputs from models that examined the independent effects of the three TFEQ constructs on trajectories of BMI changes (secondary hypothesis) are summarized in Table 3. Model 1 found a significant influence only for income (β = −0.64 change/month, 95% CI: −1.25, −0.03) and GWG (β = −0.08 change/month, 95% CI: −0.12, −0.03). Model 2 found that each construct interacted with time and that the significant interactions were independent of both prepregnancy BMI and infant feeding modality groups. Dietary restraint, disinhibition, and susceptibility to hunger were each independently associated with the linear rate of BMI change during the postpartum period. We found higher dietary restraint (−0.04 percent BMI change/month, 95% CI: −0.09, −0.00) and higher disinhibition (−0.05 percent BMI change/month, 95% CI: −0.10, −0.00) scores led to greater linear rates of BMI loss, whereas higher susceptibility to hunger scores led to greater rates of BMI gain (0.10 percent BMI change/month, 95% CI: 0.02, 0.18).

Model 3 revealed significant interactions of each construct with the polynomial factor of time (time^2^). Dietary restraint, disinhibition, and susceptibility to hunger were each independently associated with the quadratic rate of percent BMI change. To depict graphically the findings from the model, for each construct we categorized women into high and low groups by median splits (dietary restraint: low < 5, high ≥ 5; disinhibition: low < 4, high ≥ 4; susceptibility to hunger: low < 3, high ≥ 3).

Figure 3 illustrates how each construct was associated with the percent change in BMI at 4.5 months, 7.5 months, and 1 y. Overall, women with high dietary restraint (Figure 3A), high disinhibition (Figure 3B), or low susceptibility to hunger (Figure 3C) at baseline had greater percent BMI loss. By 1 y, women in the high groups for dietary restraint and disinhibition and the low group for susceptibility to hunger had lost 4.4%, 3.9%, and 5.2%, respectively, of BMI relative to baseline, whereas those in the low groups lost 1.5%, 2.0%, and 1.2%, respectively.

## 4. Discussion

A woman’s body weight status prior to becoming pregnant, and whether or not she breastfed her infant, had independent and interactive effects on both the direction and speed of weight change during the first year after giving birth. Women with prepregnancy overweight lost less weight and at slower rates than did women with prepregnancy healthy weight (1.3% vs. 7.9% BMI loss), while those with prepregnancy obesity gained weight (3.4% BMI gain). Mothers of infants who exclusively formula fed lost less (0.6% BMI decrease) and at slower rates than did mothers who breastfed their infants (5.8% BMI decrease), regardless of prepregnancy BMI. Our findings of greater loss among lactating than non-lactating mothers are consistent with prior research [6,7,8,25,26].

The novel findings in the present study come from the significant interaction between the infant feeding modality and prepregnancy BMI. While the linear rate of loss was significantly faster for lactating women with a healthy weight prepregnancy and tended to be faster for those with obesity prepregnancy, in both groups the BMI trajectories of mothers converged at 1 year. That is, at 1 year postpartum, regardless of whether they lactated or not, mothers with prepregnancy healthy weight had 7.4% BMI losses, whereas those with prepregnancy obesity had 3.5% BMI gains, relative to the start of the postpartum period. In contrast, we found striking differences in the rate, direction, and amount of BMI changes among mothers who were with overweight prepregnancy. By 1 year, non-lactating women had gained weight (2.1% BMI gain) whereas lactating women had lost weight (4.1% BMI loss). Further, nearly half of the non-lactating women, but less than one-tenth of the lactating women, were 3 or more BMI units heavier at 1 year than their prepregnancy BMI—a gain of this amount is considered to be substantial [27]. Prior research reported that postpartum weight loss was associated with lactation lasting for at least 6 months [8]. In the present study, the Lactating group exclusively breastfed for at least 4.5 months, with the majority (80%) breastfeeding for 10 months or more, a substantially longer duration.

Three hypotheses, not mutually exclusive, may explain the benefit of lactation on BMI trajectories for women with prepregnancy healthy weight or overweight and why its benefit was attenuated for those with prepregnancy obesity. First, the lactating mother’s body adapts to its energetic demands by changing metabolic rate and mobilizing energy stores [3]. While energy requirements are influenced by the volume and composition of the milk produced and the duration of lactation, the energy cost to support lactation is approximately 500 kcal/d for women whose infants exclusively breastfeed for 6 months [28]. Contrary to prior reports [29,30], we found no differences in the duration of lactation based on maternal prepregnancy BMI. However, an inclusion criterion for the longitudinal trial in which they participated established breastfeeding at enrolment and the intention to breastfeed for at least 4.5 months, Thus, the present group of lactating mothers were a highly motivated group and most, regardless of prepregnancy BMI, breastfed their infants for more than 10 months.

Second, the metabolic and hormonal (e.g., hyperinsulinemia) consequences of obesity may attenuate the energetic demands of lactation [31]. As determined by dual-energy x-ray absorptiometry, women with prepregnancy obesity had greater fat mass gains during each pregnancy trimester, which in turn positively correlated with GWG and postpartum fat mass retention [32]. This suggests that it is harder for those with obesity to lose weight during the postpartum period even when they breastfeed their infant.

Third, often in studies the mothers who exclusively breastfeed their infants differ from mothers who exclusively formula feed their infants in GWG and socioeconomic factors such as income, both of which were associated with the trajectories of postpartum BMI changes. In the present study, among those with either prepregnancy healthy weight or prepregnancy obesity, GWG, an established predictor of postpartum weight retention [8,33], was negatively associated with BMI changes, whereas among those with prepregnancy overweight, household income, which was related to both education and race, was negatively associated with BMI changes. Women with lower incomes often gain more weight than recommended during gestation, and they and their children are at greater risk for adverse health outcomes [34]. Taken together, these variables reflect differences in structural and social determinants of health, such as access to certain foods and health services, which in turn may impact the complex phenotype of a woman’s weight status and her management of weight during and after pregnancy [18].

The present analyses also revealed that, regardless of prepregnancy BMI or whether they lactated or not, individual differences in dietary restraint, disinhibition, and susceptibility to hunger interacted with time and were associated with different BMI trajectories. Because we measured these constructs early in the postpartum period, these scores were not the consequences but, rather, predictors of the BMI change we studied. Postpartum women who scored high in dietary restraint (e.g., consciously controlled or restricted what they ate), high in disinhibition (e.g., reported more uncontrolled eating), or low in susceptibility to hunger (e.g., less inclined to eat in response to feelings of hunger) experienced greater and faster BMI loss over time. These findings appear to be in conflict with the general population, where higher disinhibition scores are often associated with higher BMIs and greater regains following weight loss [35,36]. However, high levels of dietary restraint can attenuate the weight gain associated with concurrent high levels of disinhibition [37,38,39]—such a conscious restriction of intake can moderate the weight gain from uncontrolled eating or disinhibition. Whether the women who experienced greater rates of weight loss had higher levels of both disinhibition and dietary restraint remains unknown because we lacked the power to group the mothers based on construct combinations. This is an important area for future research.

This study was not without other limitations. We used self-reported GWG data and determined each woman’s prepregnancy BMI from a measured height but self-reported body weight. The use of self-reported body weight has been regarded as sufficient to classify women into BMI categories [20], as changes in BMI category may be less subject to bias. Further, pregnancy BMI highly correlated with BMI at baseline, the latter of which was derived from both measured weight and measured height, which suggests that their recalled prepregnancy body weights were reliable [2]. The BMIs of the women with prepregnancy obesity in the present study ranged from 30 to 59, but the sample size did not allow us to stratify women by class of obesity, as has been recently recommended [18].

## 5. Conclusions

There are myriad advantages to breastfeeding for the mother–infant dyad [40]. By studying two groups of women who differed in how they fed their infants and with no overlap in feeding modality, we found that lactation had benefits in the initial rates of the BMI trajectory across all prepregnancy BMI groups, but its benefits were most pronounced among those with prepregnancy overweight. Individual differences in the psychological eating behavior traits of dietary restraint, disinhibition, and susceptibility to hunger predicted the rate and direction of BMI changes during the first year postpartum. Of interest, outcomes of recent behavioral intervention trials suggest that these constructs hold promise as key modifiable targets for weight management during and after pregnancy [23,41].

## Figures and Tables

**Figure 1 nutrients-15-01364-f001:**
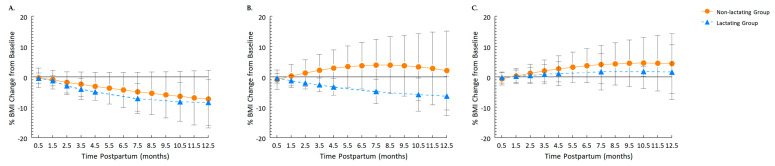
The trajectory of percent BMI change from baseline (0.5 month) to 1 year by infant feeding modality (Lactating group, Non-lactating group) for the (**A**) Prepregnancy Healthy Weight, (**B**) Prepregnancy Overweight, or (**C**) Prepregnancy Obesity group (*n* = 206 total). The linear and quadratic coefficients of the BMI trajectories from the linear mixed-effect models revealed that in both the Prepregnancy Healthy Weight and Prepregnancy Overweight groups, non-lactating women significantly differed from lactating women. The Prepregnancy Obesity group showed a tendency for the linear coefficient of the trajectories to differ between non-lactating and lactating women, but there was no significant difference in the quadratic coefficient.

**Figure 2 nutrients-15-01364-f002:**
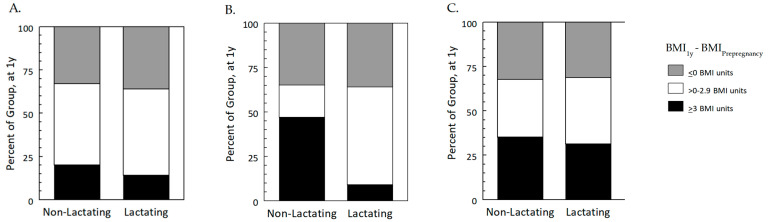
Percentage of lactating and non-lactating women who at 1 year postpartum (*n* = 158) were the same BMI or less (< 25th percentile), 0.1–2.9 BMI units higher (25–75th percentile), or ≥3 BMI units higher (>75th percentile) than prepregnancy, by prepregnancy (PPG) BMI group: (**A**), Prepregnancy Healthy Weight; (**B**), Prepregnancy Overweight; (**C**), Prepregnancy Obesity. In the Prepregnancy Overweight group (**B**), a greater percentage of non-lactating than lactating women were in the upper quartile, meaning they were ≥3 BMI units heavier than prepregnancy.

**Figure 3 nutrients-15-01364-f003:**
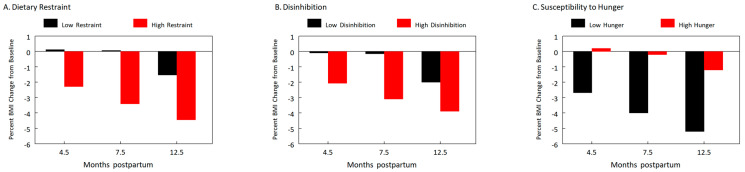
Psychological constructs of dietary restraint (**A**), disinhibition (**B**), and susceptibility to hunger (**C**) measured at baseline (0.5 month postpartum) interacted with time and were independently associated with BMI changes postpartum (*n* = 201). Each psychological construct is categorized by median split (dietary restraint: low < 5, high ≥ 5; disinhibition: low < 4, high ≥ 4; susceptibility to hunger: low < 3, high ≥ 3) and plotted as percent BMI change at 4.5 months, 7.5 months, and 1 y.

**Table 1 nutrients-15-01364-t001:** Baseline characteristics of the study participants by infant feeding modality groups.

Characteristics	Infant Feeding Modality Groups ^1^		*p*-Value ^2^
	Lactating (*n* = 96)	Non-Lactating (*n* = 112)	
Age, years	30.8 ± 5.3	27.0 ± 5.6	<0.001
Self-reported race			<0.001
Asian	2.0% (2)	0% (0)	
Black	36.5% (35) ^a^	67.9% (76) ^b^	
More than one	5.2% (5)	3.5% (4)	
White	56.3% (54) ^a^	28.6% (32) ^b^	
Household income ^3^			<0.001
<USD 35,000	40.6% (39) ^a^	73.6% (81) ^b^	
USD 35,000–USD 75,000	21.9% (21)	10.9% (12)	
>USD 75,000	37.5% (36) ^a^	15.5% (17) ^b^	
Education			<0.001
Primary school	4.2% (4)	15.2% (17)	
High/technical school	33.3% (32) ^a^	60.7% (68) ^b^	
College degree or higher	62.5% (60) ^a^	24.1% (27) ^b^	
Parity, primiparous	34.4% (33)	19.6% (22)	0.02
Prepregnancy weight (kg)	70.9 ± 18.9	78.9 ± 25.8	0.01
Prepregnancy BMI	26.0 ± 6.8	29.2 ± 8.4	0.003
Prepregnancy BMI groups			0.005
Healthy weight	60.4% (58) ^a^	38.4% (43) ^b^	
Overweight	15.6% (15)	20.5% (23)	
Obesity	24.0% (23)	41.1% (46)	
GWG, kg	14.1 ± 5.7	13.2 ± 9.2	0.42
GWG relative to IOM/NRC recommendations			0.006
Below	14.6% (14)	22.3% (25)	
Within	40.6% (39) ^a^	20.5% (23) ^b^	
Exceeded	44.8% (43)	57.1% (64)	

Values are % (*n*) or mean ± standard deviation. ^1^ *n* = 208, unless otherwise indicated. ^2^
*p*-values for main effect; ^a,b^ represents subgroups that are significantly different from each other, *p* < 0.05. ^3^ Household income, *n* = 206. Abbreviations: BMI, body mass index; GWG, gestational weight gain; IOM/NRC, Institute of Medicine and National Research Council.

**Table 2 nutrients-15-01364-t002:** Outputs from the linear mixed-effects models on influence of infant feeding modality on the trajectory of percent BMI change during the 1 year postpartum, by prepregnancy BMI category ^1^.

Characteristics	Coefficient (95% CI) ^1^
Prepregnancy Healthy Weight group	
Age	−0.01 (−0.14, 0.12)
Income	−0.25 (−1.04, 0.54)
GWG	−0.14 (−0.24, −0.04)
Time	−1.37 (−1.67, −1.07)
Time^2^	0.06 (0.04, 0.08)
Infant feeding group	
Non-lactating group	−0.17 (−1.47, 1.14)
Infant feeding modality group × time	
Non-lactating group	0.63 (0.19, 1.06)
Infant feeding modality group × time^2^	
Non-lactating group	−0.05 (−0.07, −0.02)
Prepregnancy Overweight group	
Age	−0.07 (−0.22, 0.09)
Income	−1.56 (−3.0, −0.16)
GWG	−0.04 (−0.20, 0.12)
Time	−0.81 (−1.56, −0.07)
Time^2^	0.03 (−0.02, 0.08)
Infant feeding modality group	
Non-lactating group	−1.27 (−3.66, 1.12)
Infant feeding modality group × time	
Non-lactating group	2.10 (1.16, 3.03)
Infant feeding modality group × time^2^	
Non-lactating group	−0.11 (−0.17, −0.06)
Prepregnancy Obesity group	
Age	0.06 (−0.09, 0.21)
Income	−1.03 (−2.11, 0.05)
GWG	−0.07 (−0.13, −0.01)
Time	0.43 (−0.11, 0.97)
Time^2^	−0.02 (−0.06, 0.01)
Infant feeding modality	
Non-lactating group	−0.58 (−2.01, 0.86)
Infant feeding modality group × time	
Non-lactating group	0.60 (−0.03, 1.23)
Infant feeding modality group × time^2^	
Non-lactating group	−0.03 (−0.06, 0.01)

^1^ Output from linear mixed-effects models to determine the main effect of infant feeding modality, age, income, and GWG, and interactions with time and time^2^ for each prepregnancy BMI group. Lactating women were the reference group for the model. Abbreviations: BMI, body mass index; GWG, gestational weight gain.

**Table 3 nutrients-15-01364-t003:** Coefficients from mixed-linear effects models on the influence of psychological eating behaviors of women on the trajectory of BMI changes across first year postpartum ^1^.

Characteristics	Model 1	Model 2	Model 3
	Coefficient (95% CI)	Coefficient (95% CI)	Coefficient (95% CI)
Age	0.01 (−0.08, 0.09)	0.01 (−0.08, 0.09)	0.01 (−0.08, 0.09)
Income	−0.64 (−1.25, −0.03)	−0.64 (−1.25, −0.03)	−0.64 (−1.25, −0.03)
GWG	−0.08 (−0.12, −0.03)	−0.08 (−0.13, −0.03)	−0.08 (−0.13, −0.02)
Infant feeding modality group	0.38 (−0.50, 1.27)	0.38 (−0.50, 1.27)	0.38 (−0.51, 1.26)
Prepregnancy BMI group	0.18 (−0.38, 0.71)	0.17 (−0.38, 0.71)	0.17 (−0.37, 0.72)
Dietary restraint	−0.05 (−0.18, 0.09)	−0.05 (−0.18, 0.09)	0.10 (−0.05, 0.24)
Disinhibition	−0.06 (−0.20, 0.09)	−0.06 (−0.21, 0.09)	0.18 (0.01, 0.33)
Susceptibility to hunger	0.10 (−0.15, 0.34)	0.09 (−0.15, 0.34)	−0.18 (−0.45, 0.08)
Time	−0.12 (−0.29, 0.06)	0.06 (−0.24, 0.36)	0.73 (0.37, 1.09)
Time^2^	−0.01 (−0.02, 0.00)	−0.01 (−0.02, 0.00)	−0.07 (−0.09, −0.05)
Dietary restraint × time		−0.04 (−0.09, −0.00)	−0.14 (−0.19, −0.08)
Disinhibition × time		−0.05 (−0.10, −0.00)	−0.20 (−0.26, −0.14)
Susceptibility to hunger × time		0.10 (0.02, 0.18)	0.28 (0.18, 0.38)
Dietary restraint × time^2^			0.01 (0.00, 0.01)
Disinhibition × time^2^			0.01 (0.01, 0.02)
Susceptibility to hunger × time^2^			−0.02 (−0.02, −0.01)

^1^ Output from stepwise linear-mixed effects models to determine the influence of the three TFEQ eating behavior constructs (i.e., restraint, disinhibition, susceptibility to hunger) on percent BMI change over 1 y. Model 1 includes the main effects of eating behavior constructs, prepregnancy BMI, infant feeding modality age, income, and GWG; Model 2 includes interactions with time; and Model 3 includes interactions with time^2^. Lactating women are the reference group for the models. Abbreviations: BMI: body mass index; GWG: gestational weight gain; TFEQ, Three-Factor Eating Questionnaire.

## Data Availability

Data described in the manuscript, codebook, and analytic code will be made available upon request pending application and approval.

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
