# Peer review of "Factors Affecting BMI Changes in Mothers during the First Year Postpartum"

_nutrients, 2023, doi:10.3390/nu15061364_

Round 1
Reviewer 1 Report
I thanks the opportunity to read this paper that I considered very interesting.
I have no considerations about the paper. Everything is very clear.
I only could found two small issues:
- In the 5th line of introduction we find “ENREF_4”. What this means?
- Page 5, first line of results description is not complete.
I considered that this paper could be published after these small corrections.
Author Response
Thank you for the reviews of our manuscript and the opportunity to submit a revision. In what follows, we provide details, in a point-by-point fashion along with the specification of page numbers, on how we addressed each query and critique. Please note that we left-justified all table and figure legends and have updated the ENDNOTE reference manager database which is embedded in the revised manuscript
Reviewer 1
1) In the 5th line of introduction we find “ENREF_4”. What this means?
Response: Thank you for pointing out that error. It has been corrected and the text has been deleted.
2) Page 5, first line of results description is not complete.
Response: The formatting has been corrected and the sentence that was not complete has been has been updated

Reviewer 2 Report
This is a very well written and comprehensive analysis of BMI trajectories postpartum and the factors influencing BMI change, including feeding modality. Some minor comments below:
Introduction
1. Error behind ref #3 = “ENREF 4”
Methods
1. It would be useful to have the years of data collection from the original studies in section 2.1.
2. I would like to see a statement in section 2.1 about whether consent was given in the original studies for unspecified data use?
3. Section 2.4, starts as “The secondary hypothesis…”. At the end of the introduction this was referred to as the “secondary goal” with no hypothesis made so this wording should be changed.
4. Section 2.4, the first sentence needs edited as it currently does not make sense ”…..with infant feeding modality determined the trajectories of postpartum BMI changes”.
5. Section 2.4, paragraph 3 also states the “secondary hypothesis”.
Results
1. There is an error in the first line of section 3.1 where the remainder of the sentence is directly under Table 1 and incorrect capitalization of the word “for”.
2. Table 1, footnote a,b needs rewording as it does not make sense – I believe the word “different” needs replaced with “difference”?
Discussion
1. Paragraph 6 starting “The present analyses…” on the 3rd line down please remove “in” from between “…were associated with different” and “…BMI trajectories”.
Supplemental Table 1
1. In the footnote you have an abbreviation for SEM but this is not used within the table?
Author Response
Thank you for the reviews of our manuscript and the opportunity to submit a revision. In what follows, we provide details, in a point-by-point fashion along with the specification of page numbers, on how we addressed each query and critique. Please note that we left-justified all table and figure legends and have updated the ENDNOTE reference manager database which is embedded in the revised manuscript.
Reviewer 2
Introduction
1) Error behind ref #3 = “ENREF 4”
Response: Thank you for pointing out that error from the reference manager system. It has been corrected and the text has been deleted
Methods
2) It would be useful to have the years of data collection from the original studies in section 2.1.
Response: The years of data collection have been added to the Appendix Figure A1 and in first paragraph in section 2.1.
3) I would like to see a statement in section 2.1 about whether consent was given in the original studies for unspecified data use?
Response: The mothers gave consent for study participation and were informed of all the measures (including anthropometry, demographics, feeding style) that we were going to collect from her and her child during the course of the trial. They were also informed that all data will be decoded and then de-identified. Data were not used in an unspecified manner.
4) Section 2.4, starts as “The secondary hypothesis…”. At the end of the introduction this was referred to as the “secondary goal” with no hypothesis made so this wording should be changed.
Response: The text has been modified from “secondary hypothesis” to “secondary goal” for consistency.
5) Section 2.5, the first sentence needs edited as it currently does not make sense ”…..with infant feeding modality determined the trajectories of postpartum BMI changes”.
Response: The first sentence has been edited and now reads, “We took several approaches to test the primary hypothesis that the mother’s body weight status prior to pregnancy would interact with whether or not she lactated in predicting the direction and trajectories of BMI changes during the first year postpartum.”
6) Section 2.4, paragraph 3 also states the “secondary hypothesis”.
Response: The text has been updated from “secondary hypothesis” to now read as “secondary goal”.
Results
7) There is an error in the first line of section 3.1 where the remainder of the sentence is directly under Table 1 and incorrect capitalization of the word “for”.
Response: The formatting has been corrected and the sentence edited.
8) Table 1, footnote a,b needs rewording as it does not make sense – I believe the word “different” needs replaced with “difference”?
Response: The wording for footnote 2 in Table 1 has been updated to signify that the superscripts a,b represent subgroups that are significantly different from each other, p<0.05.
Discussion
9) Paragraph 6 starting “The present analyses…” on the 3rd line down please remove “in” from between “…were associated with different” and “…BMI trajectories”.
Response: The word “in” has been removed.
Supplemental Table 1
10) In the footnote you have an abbreviation for SEM but this is not used within the table?
Response: The abbreviation SEM has been deleted as it is not used in the Supplemental Table 1.
